# Defining the Molecular Mechanisms of the Relaxant Action of Adiponectin on Murine Gastric Fundus Smooth Muscle: Potential Translational Perspectives on Eating Disorder Management

**DOI:** 10.3390/ijms24021082

**Published:** 2023-01-05

**Authors:** Rachele Garella, Emanuele Cassioli, Flaminia Chellini, Alessia Tani, Eleonora Rossi, Eglantina Idrizaj, Daniele Guasti, Paolo Comeglio, Francesco Palmieri, Martina Parigi, Linda Vignozzi, Maria Caterina Baccari, Valdo Ricca, Chiara Sassoli, Giovanni Castellini, Roberta Squecco

**Affiliations:** 1Department of Experimental and Clinical Medicine, Section of Physiological Sciences, University of Florence, 50134 Florence, Italy; 2Department of Health Sciences, Psychiatry Unit, University of Florence, 50134 Firenze, Italy; 3Department of Experimental and Clinical Medicine, Section of Anatomy and Histology, Imaging Platform, University of Florence, 50134 Florence, Italy; 4Department of Experimental Clinical and Biomedical Sciences “Mario Serio”, University of Florence, Viale Pieraccini, 6, 50139 Florence, Italy

**Keywords:** adiponectin, eating disorders, molecular mechanisms, biomarker, cell signaling, electrophysiological records, stomach, gap junctions, morphology, guanylate cyclase

## Abstract

Adiponectin (ADPN), a hormone produced by adipose tissue, facilitates gastric relaxation and can be a satiety signal in the network connecting peripheral organs and the central nervous system for feeding behavior control. Here, we performed preclinical research by morpho-functional analyses on murine gastric fundus smooth muscle to add insights into the molecular mechanisms underpinning ADPN action. Moreover, we conducted a clinical study to evaluate the potential use of ADPN as a biomarker for eating disorders (ED) based on the demonstrated gastric alterations and hormone level fluctuations that are often associated with ED. The clinical study recruited patients with ED and healthy controls who underwent blood draws for ADPN dosage and psychopathology evaluation tests. The findings of this basic research support the ADPN relaxant action, as indicated by the smooth muscle cell membrane pro-relaxant effects, with mild modifications of contractile apparatus and slight inhibitory effects on gap junctions. All of these actions engaged the ADPN/nitric oxide/guanylate cyclase pathway. The clinical data failed to unravel a correlation between ADPN levels and the considered ED, thus negating the potential use of ADPN as a valid biomarker for ED management for the moment. Nevertheless, this adipokine can modulate physiological eating behavior, and its effects deserve further investigation.

## 1. Introduction

White adipose tissue is considered an endocrine organ for its ability to secrete several bioactive molecules named adipokines. Among these, adiponectin (ADPN) is a pleiotropic peptide hormone that displays many different biological functions [1]. Aside from its insulin-sensitizing, anti-inflammatory, anti-apoptotic, anti-atherosclerotic, and cardio-protective functions, ADPN regulates energy homeostasis and has been recently proposed to have a role in the modulation of the hunger–satiety cycle, both at central and peripheral levels [2,3,4,5]. In fact, a number of peripherally released peptide hormones, together with mechano- and chemo-receptor signals, produce feedback to the brain to generate sensations of hunger to increase energy intake or satiation to end energy intake [6]. This extremely complex network regulates human metabolism and ensures the maintenance of energy homeostasis in physiological conditions. However, different pathological states characterized by an altered energy balance are often accompanied by changes in adipokine expression or plasma levels that usually vary in a direct or inverse proportion to adipose tissue content [7,8]. This observation encourages the research on adipokines and their mechanisms of action in order to recognize them as possible diagnostic or prognostic biomarkers or new therapeutics for diseases with altered energy balance. In this regard, eating disorders (ED) represent multifactorial pathologies often accompanied by fat mass content variation, hormone level fluctuations [7,9], and gastric alterations [10] that can affect the hunger–satiety cycle. Notably, ADPN plasma levels usually rise with weight loss or diet restriction in human subjects [11]. Recent findings obtained in mice suggested that ADPN could actually take part in the network connecting peripheral organs and the central nervous system controlling feeding behavior, representing a satiety signal. This may also be due to its ability to decrease cell excitability and contractile responses of the gastric fundus smooth muscle [5]. The gastric fundus has a main storage function, accommodating ingested nutrients by receptive relaxation through the activation of tension-sensitive gastric mechanoreceptors. This accommodation reflex spontaneously takes place under physiological conditions, and distension of the proximal stomach is regarded as a peripheral satiety signal. However, in several diseases, such as functional dyspepsia and related disorders, impaired gastric accommodation occurs [12,13]. The accommodation reflex is controlled by a vagovagal non-adrenergic, non-cholinergic pathway, including a key role of nitrergic neurons [14,15,16,17] that express neuronal nitric oxide synthase (nNOS) and produce NO. NO is a small molecule that can easily diffuse into the neighboring smooth muscle cells (SMCs) and is considered a major inhibitory neurotransmitter in the gastrointestinal (GI) tract [18]. The main NO intracellular effector, guanylate cyclase (GC), is expressed in various cell types of the GI tract, including interstitial cells of Cajal (ICC) and SMCs [19]. Therefore, NO enhances GC activity that leads to cyclic guanosine monophosphate (cGMP) production, which in turn causes a drop of intercellular Ca^2+^ and, eventually, smooth muscle relaxation [12]. In this regard, Idrizaj and colleagues [5] showed that ADPN facilitates the relaxation of the murine gastric fundus by a mechanism involving NO production and 5′ AMP-activated protein kinase (AMPK) pathways in neurons of the myenteric plexus. However, whether the GC pathway may also be involved in the ADPN-induced relaxation of the gastric fundus muscle layer remains to be elucidated.

On these bases, the aim of this multidisciplinary research was double: (i) to investigate further the molecular mechanisms of action of ADPN in gastric fundus relaxation in mice and (ii) to explore whether it could be considered a possible biological marker for ED. From this perspective, the present study was designed to encompass basic and clinical research. The basic research included a morpho-functional analysis to investigate the effects and mechanisms of action of ADPN on well-established ex vivo isolated mouse stomach fundus preparations [5,20]. In particular, based on the previously observed modulatory effects of ADPN on the mechanical responses and electrophysiological properties of the murine gastric fundus [21,22], we explored whether ADPN could induce structural and ultrastructural changes of SMCs, or affect gap junctional (GJ) communication by changes in connexin (Cx)43 expression and/or functionality. We also investigated the involvement of GC in ADPN/NO-mediated SMC quiescence. Next, in the clinical study, we engaged with healthy controls and patients with ED to test whether changes in blood levels of ADPN could have a significant correlation with body mass index (BMI) and ED psychopathology, as already reported for leptin, another well-known adipokine [23]. Moreover, we examined the possible role of ADPN levels during the clinical course of ED, looking for a possible relationship between BMI and specific reward-related behaviors, such as dietary restraint and compensatory exercise. The experimental design is summed up in Figure 1.

## 2. Results

### 2.1. Basic Research

#### 2.1.1. ADPN Relaxant Effects on SMCs of the Murine Gastric Fundus Involve GC Activity

The electrophysiological records on murine gastric fundus SMCs showed that ADPN (20 nM) added to the bath medium induced plasma membrane hyperpolarization, as indicated by a reduction of the resting membrane potential (RMP) value (Figure 2a), which caused a decrease in SMC excitability. In addition, it caused an increase in cell capacitance (Cm), an electrophysiological parameter related to cell surface extension (Figure 2b; Table 1), suggesting a relaxation-induced elongation of the cells. These data further confirm the ability of ADPN to influence the electrical properties of the SMC plasma membrane [22].

To deepen the knowledge of the molecular mechanisms of action of ADPN and its possible mediators, we investigated whether the observed functional changes induced in gastric SMCs (already suggested to be mediated by NO [5]) also involved a NO-dependent activation of GC [19,20]. To this aim, the selective GC inhibitor 1H-[1,2,4] oxadiazolo[4,3-α]-quinoxalin-1-one (ODQ) [20,24] was added to the bath medium, and the first set of records was made to observe its effects on the cells’ electrophysiological properties (Figure 2a,b). Pulse stimulation protocols applied 5 min after ODQ (1 µM) treatment showed no differences in RMP or Cm values with respect to the untreated controls (CTRL). Increasing the exposure time up to 30 min also caused no changes; hence, the electrophysiological results shown in the figures are related to 5 min ODQ exposure. Then, ADPN (20 nM) was added to the same bath solution in the presence of ODQ (1 µM), and stimulation was repeated after a further 15 min incubation. The ODQ + ADPN-treated SMCs showed similar RMP and Cm values as the untreated CTRL (Figure 2a,b; Table 1). 

Records made after longer ADPN exposure times (up to 60 min) also showed no differences. These results indicate that ADPN was no longer able to induce plasma membrane hyperpolarization and increase Cm when the GC activity was inhibited by ODQ. This finding suggests that GC activation and the consequently enhanced cGMP production mediate the effects of ADPN on plasma membrane electrical properties related to SMC relaxation. In this context, in accordance with its hyperpolarizing effect on the SMC plasma membrane, it was previously reported that ADPN was able to increase the amplitude of K^+^ currents [5]. This outcome was confirmed in the present experiments (Figure 3a–c,h), and to extend this finding, we tested the effects of ADPN on K^+^ currents when GC was blocked by ODQ. 

A representative experiment is reported in Figure 3d–g: ODQ alone (Figure 3e,g) had no significant effects on K^+^ currents amplitude compared to the untreated controls (Figure 3d,g). In the presence of ODQ, ADPN was no longer able to increase K^+^ current amplitude (Figure 3f,g). The related I–V plot (where the outward current amplitude was measured at the end of the step pulse) was almost coincident with that obtained in the untreated control SMCs (Figure 3g). The mean amplitudes of the currents evoked by any test pulse related to all the experiments were plotted as a function of the voltage to analyze this phenomenon better (Figure 3h). Again, these data confirm that the ability of ADPN to increase K^+^ current was dependent on GC activation.

In order to gain further insights into the cellular and molecular mechanisms underpinning the relaxant effects of ADPN on gastric SMCs, we evaluated whether the alterations of bioelectrical properties were associated with morphological changes. Light microscopic evaluation of hematoxylin and eosin-stained sections of ADPN-treated gastric fundus strips revealed histological features similar to those of the untreated CTRL strips (Figure 4a–d). In detail, we observed a typical mucosa organization with stratified squamous epithelium, a lamina propria made up of loose connective tissue, thin muscularis mucosae and submucosal stromal layers, and a thick, smooth muscle coat consisting of an inner layer with circularly arranged SMCs and an outer layer with longitudinally arranged ones. Confocal immunofluorescence analysis of the α-smooth muscle actin (sma) expression and distribution (Figure 4e–l) in the smooth muscle layers did not show substantial differences between control and ADPN-treated strips. Quantitation of green fluorescence intensity related to α-sma in circular and longitudinal muscle layers yielded the following values (arbitrary units, a.u.): control 158.88 ± 5.66, ADPN-treated 159.27 ± 4.36 (Student’s *t*-test: *p* = 0.8667, not significant). Ultrastructural analysis of the muscle layers revealed that the SMCs of ADPN-treated samples did not show substantial differences in the amount and morphology of the typical structure of contractile machinery, namely plasma membrane caveolae or sarcoplasmic reticulum profiles, as compared to untreated controls (Figure 4m–o). However, ADPN-treated cells (Figure 4n) showed a slightly looser network of the contractile filaments as compared with the untreated controls (Figure 4m), suggestive of a more relaxed state and consistent with the electrophysiological results. On the other hand, positive control samples treated with the contractile agonist carbachol (CCh) (Figure 4o) showed a denser contractile filament network as compared with the untreated controls.

Collectively, these results indicate that ADPN treatment only slightly affects gastric SMC morphological features and contractile machinery. We thus suggest that the modulatory role of this adipokine on the cell relaxant response primarily involves functional changes in plasma membrane excitability. 

#### 2.1.2. ADPN Slightly Reduces Gap Junction Function and Cx43 Expression in Gastric SMCs by GC Activation

In order to investigate whether ADPN could affect the electrical coupling between SMCs in the gastric fundus strips, the GJ currents were recorded. Trans-junctional currents (Ij) recorded in untreated SMCs showed a voltage-dependent, asymmetrical time course for positive and negative trans-junctional voltages, Vj. Representative families of Ij current traces are reported in Figure 5a. Records were repeated from the same cell at least 15 min after adding ADPN (20 nM) to the bath solution. As observed in the representative experiments shown in Figure 5b, we observed no changes in the Ij time course compared to the untreated controls. However, the Ij current amplitude was often slightly reduced in the presence of ADPN, both for the instantaneous (Ij,inst in Figure 5c) and the steady state Ij phases (Ij,ss in Figure 5d).

Collectively, the experimental data suggest that ADPN tends to decrease the Ij currents, although the differences in Ij amplitudes were not statistically significant compared with the untreated CTRL (Figure 5e,f). 

Based on Ij’s behavior that suggests the presence of voltage-dependent GJ, we analyzed the effects of ADPN on the expression of Cx43 next (the typical connexin isoform of voltage-dependent connexons). In line with the electrophysiological recordings, we observed a significant reduction of Cx43 expression in the ADPN-treated samples (Figure 5l) as compared with the untreated CTRL (Figure 5k). Although ADPN only showed a tendency to inhibit the trans-junctional currents, which did not reach statistical significance, we wanted to test if blocking GC by ODQ could negate this effect. In fact, in the presence of ODQ as well as ODQ + ADPN, the Ij amplitude remained similar to the untreated controls. The overall Ij,inst–Vj and the Ij,ss–Vj plots related to the three conditions were not statistically different from each other, confirming the complete lack of effect of ADPN in the presence of ODQ (Figure 5i,j). The data obtained from confocal immunofluorescence of the samples treated with ODQ + ADPN confirmed this finding (Figure 5n). In fact, Cx43 expression was similar and not statistically different to the untreated CTRL and, hence, significantly higher than that observed in the presence of ADPN alone (Figure 5o). ODQ treatment (Figure 5m) did not cause significant alterations in Cx43 expression compared to CTRL. Overall, the above results indicate that GC can also have a role in the modulation of GJ function by ADPN because, when it is pharmacologically inhibited, ADPN is no longer able to decrease the trans-junctional current and Cx43 expression. 

### 2.2. Clinical Study

Since it has been proposed a role for ADPN as a peripheral satiety signal originating from the stomach [25], and after having elucidated its mechanism of action, we aimed to explore the possibility of using adipokine as a biological marker for ED in a clinical setting. According to our previous study on leptin [23], plasma ADPN was measured in two age-matched cohorts consisting of 41 healthy subjects (HC) and 62 patients: thirty-four with a diagnosis of Anorexia Nervosa (AN), fourteen of Bulimia Nervosa (BN), and fourteen of Binge-Eating Disorder (BED). The intra-assay variability was 1.80% CV (coefficient of variation), whereas the overall inter-assay variability was 8.89% CV. The main demographic and clinical characteristics of the same cohorts were previously reported [23]. As expected, ED patients reported significantly higher scores than the control subjects in almost all of the administered questionnaires, except for Emotional Eating Scale (EES) depression. However, no differences in ADPN plasma levels were found among any of the diagnostic groups or between patients and controls (Table 2). No significant differences were found in age- and BMI-adjusted comparisons (F = 0.84, *p* = 0.475).

In particular, no association between ADPN plasma levels and BMI was found (β = −0.02, *p* = 0.851), nor between ADPN and indices of general and ED-specific psychopathology or pathological eating behaviors (Table 3). Given the absence of any meaningful correlation, no further statistical analyses were performed.

## 3. Discussion

The recent literature on the interplay between the central nervous system, GI tract, and adipose tissue has established the involvement of several mechanisms in the control of the hunger–satiety cycle and eating behavior [26,27]. Different endogenous signaling molecules, such as neurotransmitters, lipids, and hormones, including adipokines from the adipose tissue, have a role in the regulation of the energy balance, taking part in a complex, interlocked network. Notably, all of these signals respond to objective homeostatic factors of energy balance and also to subjective psychological ones related to rewarding behaviors [28,29,30,31,32,33,34], thus further complicating the scenario of food intake regulation. 

This study revolves around the role of ADPN in eating behavior. Our first endpoint was to add insights into the molecular mechanisms underpinning ADPN action by morpho-functional analyses on murine gastric fundus smooth muscle. This represents a necessary starting point for viewing ADPN as being involved in eating-related disorders that often show gastric alterations and hormone plasma level fluctuations. Dysregulation of peptides that are able to exert orexigenic or anorexigenic effects likely alters the correct feeding behavior, encouraging research on the possible repercussions of this alteration. Accordingly, the effect of ADPN was first investigated in mice, which represents a consolidated animal model for basic research. In particular, the choice of the gastric fundus as a peripheral organ relies on its key role in sensing meal content [35]. In fact, the presence of a meal in the stomach induces a mechanical distension reflex, and as gastric emptying proceeds, nutrients (especially lipids and proteins) stimulate the release of gut hormones. Collectively, these signals are sent to the brain to modulate appetite and food intake. Recent studies in humans provided significant insights into how the detection of intraluminal, meal-related stimuli can quickly modulate appetite and energy intake [36] and how abnormal sensing of such stimuli can be associated with several eating-related disorders, such as functional dyspepsia, obesity, anorexia of aging. In addition, altered blood levels of adipokines are associated with several clinical forms of ED [10], underpinning the need for identifying new diagnostic/prognostic biomarkers and therapeutic strategies. However, the mechanisms underlying these dysregulations still need to be better understood before any translations to clinical practice.

The basic research results of the present study clearly support a pro-relaxant action of ADPN on the SMCs of the murine gastric fundus, suggesting its role as a peripheral satiety/anorexigenic signal. Earlier studies showed that ADPN was able to modulate murine gastric fundus relaxation by decreasing the mechanical activity [21] and influencing the electrophysiological properties of gastric SMCs [22]. GI motility is a very complex phenomenon resulting from the dynamic interaction of muscular, neuronal, and endocrine tissues, each with different structural and functional properties. In particular, gastric motility is finely regulated by interactions among the intrinsic spontaneous pacemaker activity due to ICC, as well as the electrically coupled SMCs of the muscle layer, enteric nervous system, and the neural network located within the gastric wall, and a variety of hormonal stimuli. At the muscular level, depolarization of the SMC membrane is rapidly transmitted to the neighboring SMCs by GJ-mediated electrical coupling, thus activating the whole muscle layer [37]. Cx43, the main widespread protein forming the gap junctions responsible for electrical propagation, shows voltage-dependent activation. The present electrophysiological records confirm that ADPN induces hyperpolarization of the plasma membrane of gastric SMCs, thereby reducing their excitability and functional activation. A similar mechanism of action was observed in response to a high-sodium diet in rat aortic rings, where hyperpolarization induced by the NO/GC pathway counteracted the sustained contraction [38]. Further confirmation of ADPN pro-relaxant ability comes from the observed increase in cell capacitance, the electrophysiological parameter related to cell surface extension. This finding suggests that elongation of SMC has occurred, although no obvious changes in cell shape were clearly detected by conventional histology. On the other hand, ultrastructural observation detected mild effects of ADPN in terms of loosening the contractile filament framework. This fits well with the electrophysiological experiments by indicating an overall pro-relaxant effect of ADPN on gastric SMCs, induced by both a reduction of plasma membrane excitability and Cx43-mediated electrical coupling. 

To add further knowledge to the known mechanisms of action of ADPN [5], we demonstrate here that its effects also involve GC activation, as shown by the findings that the selective GC inhibitor, ODQ, abolished the pro-relaxant effects of ADPN. This ADPN/NO/GC signaling pathway leads to a rise in intracellular cGMP, which in turn can activate protein kinase G, a well-known ion channel modulator [39,40]. Similar to the inhibitory effects on GI motility of another pro-relaxant hormone, relaxin [20], the potential ADPN-induced ion channel modulation may be a key player of membrane hyperpolarization and, hence, of less effective excitation–contraction coupling. The previously reported reduction of Ca^2+^ influx through voltage-dependent Ca^2+^ channels due to ADPN [5] further supports the conclusion that membrane ion channels may be a main target of ADPN to blunt gastric smooth muscle motility. 

Although cGMP-independent relaxing effects have been demonstrated to occur in other districts, such as vessels and platelets [41], the present findings indicate that the GC/cGMP is the key mechanism of the relaxant effect of ADPN on the gastric smooth muscle. This could be a starting point to discover new therapeutics for gastric motor disorders, such as stimulators of NO-GC or phosphodiesterase (PDE) inhibitors, which could be used to increase cGMP levels [40] when a relaxant effect is needed. We also found that the same ADPN/NO/GC pathway can modulate—albeit weakly—the Cx43 expression and electrical coupling in gastric SMCs. Of note, a functional association between GC and Cx43 was also reported in the heart, leading us to hypothesize that the NO/GC/Cx43 pathway may be a protective mechanism against stress-induced arrhythmia [42]. Assuming that GJ impairment may concur with the pathogenesis of GI functional diseases, molecules able to enhance GJ communication deserve to be investigated as potential new drugs.

To sum up, the results of our study highlight the role of ADPN as an inhibitor of gastric smooth muscle excitability, which, while facilitating relaxation, may serve as an additional satiety signal acting at the peripheral level. Interestingly, ADPN is currently viewed as a likely regulator of hunger and satiety, even though most research on this subject has been performed in rodents and has focused on the central nervous system, particularly the hypothalamus [43]. Undoubtedly, the mechanisms through which ADPN modulates food intake are complex and multifactorial and are often associated with feeding status, glucose content in the cerebrospinal fluid, and fat mass content. In the present study, we attempted to combine the information obtained from the ex vivo rodent model with that from a clinical study on human subjects in order to investigate the potential role of ADPN in the psychopathological condition of ED, where blood levels are often altered [44,45], and gastric motility is affected [46,47,48]. It should be noticed that the development of ED is due to the multiplex contribution of several factors that influence eating behavior. These include early life adverse conditions, genetic predisposition, and a number of biological factors, such as neurotransmitters and peripheral hormones, that may act directly on the central nervous system or indirectly on peripheral organs. In this scenario, ADPN expression and plasma levels have been shown to change in relation to feeding status (both upon food intake or deprivation [43]) and have been shown to fluctuate in ED [49], giving rise to controversy about its actual role. Since the results obtained from the basic research study in mice supported the idea that ADPN could be an anorexigenic signal and laid the foundation for translational studies on human subjects, we were prompted to speculate that it may also have a role in pathological feeding behaviors. However, the clinical study gave inconclusive results at the moment. In fact, the ADPN plasma levels in patients with AN did not correlate with BMI, nor were differences detected with respect to the healthy controls. Similar negative results have already been reported [49].

### Study Limitations and Future Perspectives

Our preclinical study was based on mice only, which are commonly considered valuable and very standardized models for the study of basic biological processes. However, concerning the mouse model, one of the main limitations was the choice of deliberately healthy animals to achieve the ex vivo analyses. Nonetheless, this is an obligatory starting point to study the molecular mechanisms of action of ADPN on the stomach since, to our knowledge, this kind of ADPN effect on peripheral aspects of the hunger–satiety cycle’s control in physiological conditions was lacking. Future studies could be repeated on different species of rodents to verify their reliability. The use of human biopsies could be opportune, but this kind of sample is very hard to obtain in large numbers, especially from healthy donors, to achieve a powerful statistical analysis. With regard to rodent models, in future studies, it would be appropriate to consider animal models of ED [50,51] to evaluate eventual dysregulated mechanisms and try to better predict the behavior in patients with ED. Finally, animal models could hopefully be used to perform in vivo studies. 

Concerning the clinical study, a first limitation was the examination of a population of patients suffering from AN, BN, and BED, which are the most common but are not all EDs. Therefore, we should recognize that statistical significance testing in the report of the present study may give only a partial representation of the question. However, the analysis of the association between ADPN and ED-related psychopathology and behaviors shown in a dimensional way represents an additional strength of the study. Moreover, the cohort examined could be somehow heterogeneous, and in any case (or this is the reason), there is currently no clear consensus about the relationship between ADPN plasma levels and BMI. This is mainly due to the fact that ADPN blood levels generally fluctuate in ED [49] and can depend on the feeding status [43]. Importantly, in AN, they can be restored by weight recovery, even with little increase in BMI [49]. Since it has been observed that subjects having a thin constitution have the predisposition to show higher ADPN levels related to their lower BMI, nutrition can indeed determine the reduction of this adipokine as they otherwise might have shown hyperadiponectinemia, according to their BMI [49]. A further limitation of this study could be that, although all analyses on clinical samples were BMI-adjusted, the evaluation of BMI may not reflect the actual fat mass quantity and, therefore, the actual amount of adipose tissue able to produce ADPN. Some other techniques, such as the bioelectrical impedance analysis of body fat percentage, could be used for a more reliable evaluation of body fat mass [52]. Even if the analysis of our results is limited by the cross-sectional nature of the research design, they offer a preliminary background for further studies that try to find a correlation between adipokine plasma level oscillations following BMI alterations and the maintenance of pathological ED behaviors. In this regard, a larger (as well as a longitudinal) study could be useful in bringing additional data to demonstrate the reliability of the outcomes of the present research. The analysis is in progress in our laboratories to collect data for this purpose, and we are aware that the actual probability that a research claim is true depends on a number of factors, such as study bias, power, and the number of other studies on the same question [53]. 

Lastly, although combining an ex vivo preclinical study on an animal model with a clinical study based on blood samples from human subjects may seem like a big leap, this decision was dictated in the early stages by the need to perform the least invasive analysis possible. Since the results obtained on ADPN plasma levels seem to not show any significant correlation with the investigated ED, we may think that, for the moment, it is not worth repeating further morpho-functional investigation of ADPN effects on human gastric samples. 

## 4. Conclusions

In conclusion, our present preclinical results add novelty to the molecular mechanisms by which ADPN exerts a pro-relaxant effect on gastric smooth muscle as a whole and on individual SMCs. On the other hand, data obtained in the clinical study did not reveal statistically meaningful correlations between ADPN plasma levels and any of the investigated clinical parameters in patients suffering from AN, BN, and BED. However, with the limitations discussed above, our results seem to support a role for ADPN in eating behavior only in physiological conditions while negating the working hypothesis that ADPN may be a valid, exploitable biomarker for the management of the EDs considered in our study. For this purpose, future studies will conceivably focus on investigating other biological factors known to be involved in the regulation of feeding. Nonetheless, a thorough understanding of the molecular mechanisms of action of ADPN and activated signaling pathways can help develop new effective diagnostics and therapeutics for gastric motility disturbances/impaired gastric accommodations.

## 5. Materials and Methods

### 5.1. Basic Research: Studies on the Isolated Murine Gastric Fundus 

#### 5.1.1. Animals, Sample Preparations, and Treatments

We performed experiments on 8- to 12-week-old female mice (C57BL/6; Charles River, Lecco, Italy), as previously reported [5,20]. The mice were kept at a controlled temperature (21 ± 1 °C) under a 12 h light/12 h dark photoperiod and nourished with ordinary laboratory food and water. The mice were killed by a rapid cervical dislocation to lessen animal pain. The stomach was immediately extracted from the abdomen, and from the dissected gastric fundus, 2–3 full-thickness longitudinal strips (2 × 10 mm) were cut and subjected to the following treatments: Mouse recombinant ADPN was added to the bath medium (final concentration: 20 nM) at room temperature, and its effects were investigated at least after 15 min; ODQ (final concentration of 1 µM) to inhibit GC [20] was added to the bath solution for at least 5 min, and then we added ADPN in its concomitant presence, and the effect of GC inhibition was tested at least 15 min later; CCh (1 µM) was added to the bath medium for 1 h at room temperature to induce the muscle layer contraction. In any case, the drug concentrations were in the range of those previously proven to be effective [20,22]. Untreated strips served as CTRL.

All of the chemicals, mouse recombinant adiponectin, nifedipine, tetrodotoxin (TTX), CCh, and ODQ were purchased from Sigma-Aldrich (St. Louis, MO, USA). 

#### 5.1.2. Morphological Analysis 

##### Hematoxylin and Eosin Staining (H&E)

Untreated (CTRL) and ADPN-treated gastric fundus full-thickness specimens (*n* = 3 each) were fixed with 10% formalin in phosphate-buffered saline (PBS), dehydrated with a graded alcohol series, cleared in xylene, and embedded in paraffin. At least ten 5 µm-thick sections were cut from each sample (*n* = 30 for each experimental condition), deparaffinized, and routinely stained with H&E. Tissue morphology was then recorded under a light microscope (Leica DM4000 B) equipped with a DFC310 FX 1.4-megapixel digital color camera and software application suite LAS V3.8 (Leica Microsystems, Mannheim, Germany).

##### Transmission Electron Microscopy (TEM) 

For TEM analysis, the gastric fundus strips (*n* = 3) were exposed for 1 h at room temperature to CCh treatment. In parallel, other strips (*n* = 3) were treated with ADPN-added or plain (CTRL, *n* = 3) physiological bath solutions for the same time interval. Then, the mucosa and submucosa were carefully removed under a dissecting microscope in order to extract the muscle layer. CTRL, CCh-treated, and ADPN-treated gastric fundus specimens were fixed in Karnovsky’s fixative overnight at 4 °C, post-fixed in 1% OsO_4_ in 0.1 M phosphate buffer (pH 7.4) for 1 h at room temperature, dehydrated in a graded acetone series, passed through propylene oxide, and then embedded in Epon 812 (Sigma-Aldrich, Cat # 45345). Ultrathin sections (60 nm thick) were contrasted with UranyLess EM stain (Electron Microscopy Sciences, Foster City, CA, USA, Cat # 22409) and alkaline bismuth subnitrate and then examined using a Jeol 1010 electron microscope (Jeol, Tokyo, Japan) at 80 kV equipped with a Jeol Veleda high-resolution digital camera.

##### Confocal Laser Scanning Microscopy

Three gastric fundus strips for each experimental condition were processed for confocal laser scanning microscopy analyses. Paraffin-embedded gastric fundus tissue sections (8 µm thick) were essentially processed as previously reported and incubated with the following primary antibodies (overnight at 4 °C): rabbit polyclonal anti-Cx43 (1:100; Cell Signaling Technology, Danvers, MA, USA, Cat #3512S, Lot# 7, RRID AB_2294590), and mouse monoclonal anti-α-sma (1:100; Abcam, Cambridge, UK, Cat # ab7817, Lot# GR3246513, RRID AB_262054). The immunoreactions were revealed by incubation with the following specific secondary antibodies: anti-mouse or anti-rabbit Alexa Fluor 488-conjugated IgG (1:200; 1 h at room temperature, Molecular Probes, Eugene, OR, USA, Cat # A11001, A11034). Negative controls were obtained by substituting the primary antibody with non-immune serum, and the secondary antibody cross-reactivity was assessed by omitting the primary antibody. The fluorescent dye Propidium iodide (PI, 1:100, for 2 min at room temperature, Molecular Probes, Cat # P1304MP) was used to label nuclei. Observations were performed by using a confocal Leica TCS SP5 microscope (Leica Microsystems), as previously reported using a Leica Plan Apo 40xNA objective. Optical section series (1024 × 1024 pixels each, pixel size 204.3 nm, 209 × 209 μm optical square field, and 0.4 μm in thickness) were acquired at intervals of 0.6 μm and projected onto a single ‘extended focus’ image. Densitometric analysis of α-sma and Cx43 fluorescent signal intensity was performed on digitized images by using ImageJ software (Version 1.49 v, RRID:SCR_003070; NIH, Bethesda, MD, USA). In particular, five regions of interest (ROI; 25 × 25 µm) were analyzed for each confocal stack (five for each experimental condition). The experiments were performed in triplicate (*n* for each experimental point = ROI = 75). 

#### 5.1.3. Electrophysiological Recordings

According to our earlier papers [5,20,22] we performed intracellular recording by a conventional high resistance (around 60–70 MΩ) microelectrode introduced in a cell of the smooth muscle layer. Microelectrodes were immediately prepared before the recordings from borosilicate glass capillaries (GC 100-7.5; Clark, Reading, UK) by using a vertical puller (Narishige PC-10) and were filled with the following internal solution (mM): KCl 130, NaH_2_PO_4_ 10, CaCl_2_ 0.2, ethylene-bis(oxyethylenenitrilo)tetraacetic acid (EGTA) 1, MgATP 5 and 4-(2-hydroxyethyl)-1-piperazineethanesulfonic acid (HEPES)/KOH 10 (pH = 7.2). During the records, the gastric strip was constantly superfused (Pump 33, Harvard Apparatus, Holliston, MA, USA) at a rate of 1.8 mL min^−1^ with the Krebs–Henseleit physiological bath solution: 118 mM NaCl, 4.7 mM KCl, 1.2 mM MgSO_4_, 1.2 mM KH_2_PO_4_, 25 mM NaHCO_3_, 2.5 mM CaCl_2_, and 10 mM glucose (pH 7.4). RMP recordings were performed using the Krebs–Henseleit solution as the control bath medium by using the current clamp mode of our amplifier with a stimulus waveform: I = 0 pA. Still using this bath solution, we regularly evaluated the cell capacitance (Cm) in voltage clamp conditions by applying two 75 ms long step voltage pulses at −80 and −60 mV, starting from a holding potential (HP) of −70 mV. The delayed rectifier K^+^ currents were recorded in voltage-clamp mode using a modified Krebs–Henseleit solution with specific channels blockers: nifedipine (10 µM) to block L-type Ca^2+^ currents and tetrodotoxin (TTX 1 µM) to block Na^+^ currents. The K^+^ current activation was evoked by 1 s long voltage step pulses ranging from −80 to 50 mV applied in 10 mV increments (HP = −60 mV). Capacitive and any other linear voltage-independent ionic current were removed online by the P/4 procedure [5]. To record the currents flowing through GJs, we used the bipolar pulse protocol [54]. Concisely, starting from a holding potential (HP) = 0 mV, the impaled cell was stimulated by a bipolar 5 s pulse protocol starting at the trans-junctional voltage Vj = ± 10 mV up to ± 150 mV in 20 mV increments. The trans-junctional current flowing through GJs is named here as Ij. In particular, the instantaneous current amplitudes measured at the peak are termed Ij,inst (instantaneous trans-junctional current), and those measured at the end of any pulse that is at the steady state are termed Ij,ss (steady-state trans-junctional current). For a correct estimation of the test currents recorded from the cells of different sizes, any current amplitude was normalized to the cell linear capacitance Cm, where Cm is regarded as an index of the cell-surface area since the membrane-specific capacitance has a constant value of 1 µF/cm^2^.

All of the experiments were performed at room temperature (20–22 °C). The setup for electrophysiological records was detailed in previous papers [5] and comprised an Axopatch 200B amplifier (Axon Instruments, Union City, CA, USA), an analog-to-digital/digital-to-analog interface (Digidata 1200; Axon Instruments), and pClamp 6 software (Axon Instruments). 

### 5.2. Clinical Research

A cross-sectional observational study was conducted at the Psychiatric Unit of the University of Florence, Italy. The present study includes analyses performed on additional data collected as part of a previous study, where more details on all procedures can be found [23].

#### 5.2.1. Participants

The ED patients were enrolled between November 2017 and May 2019 (referred by their general practitioner or other clinicians), with the following inclusion criteria: female sex, age 18–60 years, current diagnosis of AN, BN, or BED according to DSM-5 criteria (American Psychiatric Association, 2013), absence of comorbid conditions precluding the possibility of being treated at the unit and the correct compilation of the questionnaires (e.g., schizophrenia, bipolar I disorder, acute psychotic disorder, illiteracy, intellectual disability, or severe medical comorbidities), and no current use of psychoactive medications (with the exception of antidepressants and benzodiazepines). Of the 74 ED patients referred, five subjects declined to participate, and seven were excluded (one with bipolar disorder, three with severe medical conditions, and three used antipsychotic or mood stabilizers). Healthy controls (HC) were recruited among students at the University of Florence, with the following inclusion criteria: female sex, age 18–60, absence of any mental disorder as assessed by a clinical evaluation, and BMI between 18.5 and 25.0 kg/m^2^.

#### 5.2.2. Assessment and Measures

Two expert psychiatrists (G.C. and V.R.) performed an initial psychiatric evaluation and collected sociodemographic and clinical data, including frequency of binge-eating episodes, compensatory exercise, and self-induced vomiting in the previous month. Serum blood samples for ADPN measurements were drawn at 8:00 a.m. after a night of fasting. Both patients and HCs followed the same study procedures. Serum ADPN was measured with Human Adiponectin ELISA (Enzyme-Linked Immunosorbent Assay) (ab108786) kit (Abcam) in accordance with the manufacturer’s instructions. All participants completed the following psychometric questionnaires: the Symptom Checklist-90 revised (SCL-90-R GSI) for general psychopathology, the Eating Disorder Examination Questionnaire version 6.0 (EDE-Q 6.0) for ED-specific psychopathology, and the Emotional Eating Scale (EES) for emotional eating. These instruments were described in more detail in the previous paper [23].

### 5.3. Data Analyses and Statistical Tests

#### 5.3.1. Statistical Analysis of Electrophysiological Data

The mathematical and statistical analysis was performed by pClamp6 (Axon Instruments) and Excel (Microsoft Office 2016, Microsoft Corporation, Redmond, WA, USA). The results of the experiments are expressed as mean ± standard deviation (SD). The average values of two datasets were compared by Student’s *t*-test, assuming that values follow a normal distribution, as assessed by the Shapiro–Wilk test [55]. A one-way ANOVA followed by Bonferroni’s post hoc test was used for multiple comparisons. Statistical significance was set to *p* < 0.05, and *n* represents the number of cells investigated.

#### 5.3.2. Statistical Analysis of Morphological Data

A one-way ANOVA followed by Bonferroni’s post hoc test was used for multiple comparisons, and a Student’s *t*-test was used to compare the average values of two datasets, both performed with the open-source statistical software Jamovi v.2.2.2 [56].

#### 5.3.3. Statistical Analysis of Clinical Data

Continuous variables were reported as mean ± SD. Two different approaches were adopted to investigate the possible role of ADPN as a marker of EDs in the clinical samples. First, between-group comparisons were performed in order to compare the three main ED diagnostic categories (AN, BN, and BED) and healthy control subjects. Comparisons between groups were performed by means of ANOVA and age- and BMI-adjusted Analysis of Covariance (ANCOVA), and *n* indicates the number of subjects.

Secondly, correlates of ADPN were investigated in the whole clinical sample using multiple age- and BMI-adjusted linear regression models, with psychometric and behavioral measurements as dependent variables and ADPN levels as the independent variable. While between-group comparisons allowed us to investigate whether specific ED diagnoses were characterized by different levels of ADPN, the second correlational approach allowed us to investigate the possible association of ADPN with ED-related psychopathology and pathological behaviors in a dimensional and non-categorical way, in accordance with a transdiagnostic model of ED.

All analyses were performed using R Statistical Software v.4.2.1 (R Core Team, 2022; R Foundation for Statistical Computing, Vienna, Austria) which is a language and environment for statistical computing [57].

## Figures and Tables

**Figure 1 ijms-24-01082-f001:**
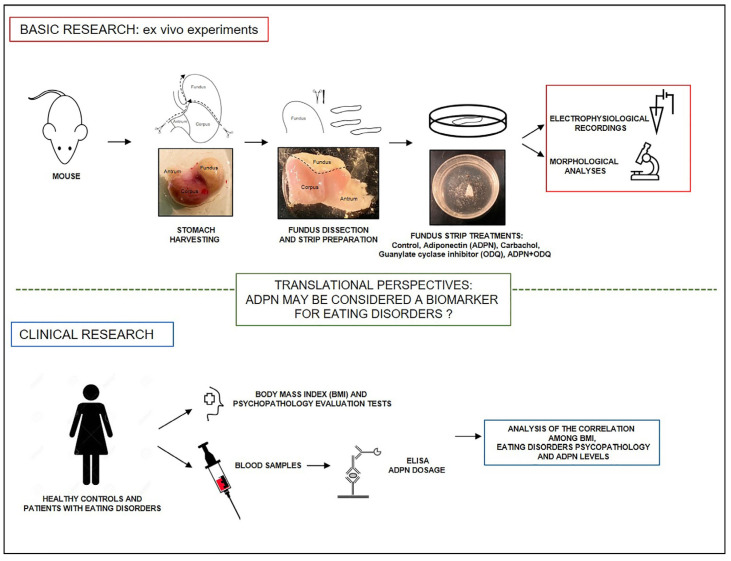
Experimental design of this study.

**Figure 2 ijms-24-01082-f002:**
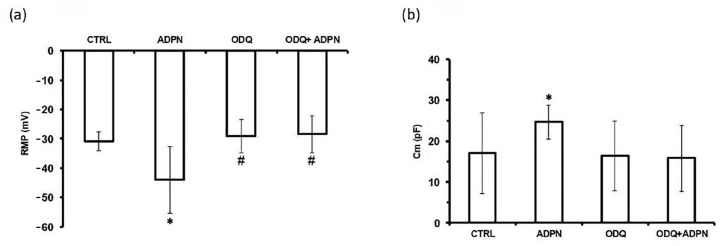
The effects of adiponectin (ADPN) on resting plasma membrane (RMP) potential and membrane capacitance (Cm) of gastric fundus smooth muscle cells (SMCs) involve guanylate cyclase (GC) activity. (**a**) RMP evaluation (in mV) in the untreated specimen (CTRL, *n* = 5), ADPN (20 nM)-treated (ADPN, *n* = 6), selective GC inhibitor 1H-[1,2,4] oxadiazolo[4,3-α]-quinoxalin-1-one (ODQ, 1 μΜ)-treated (data related to 5 min ODQ exposure; ODQ, *n* = 4), and ADPN added in the concomitant presence of ODQ and stimulation repeated after 15 min of exposure (ODQ + ADPN, *n* = 4). (**b**) Evaluation of Cm (in pF) as an index of the cell surface. Control SMCs (CTRL, *n* = 21), ADPN-treated (ADPN, *n* = 6), ODQ-treated (ODQ, *n* = 4), and ADPN-treated in the presence of ODQ (ODQ + ADPN, *n* = 5). Data are expressed as mean ± SD. *n* is the number of SMCs. * *p* < 0.05 vs. CTRL; # *p* < 0.05 vs. ADPN (one-way ANOVA with Bonferroni’s correction).

**Figure 3 ijms-24-01082-f003:**
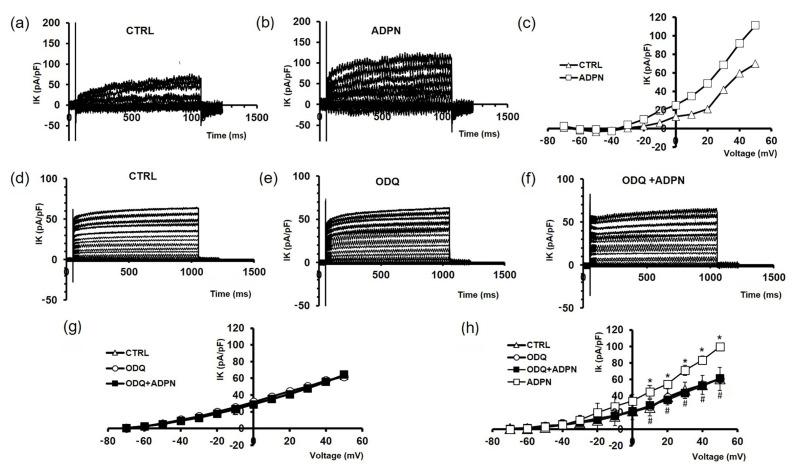
The effects of adiponectin (ADPN) on outward K^+^ currents evoked in the smooth muscle cell (SMCs) from the gastric fundus involve guanylate cyclase (GC) activity. Representative family of outward K^+^ currents evoked by voltage steps from −80 to +50 mV (holding potential, HP = −60 mV) in 10 µM-nifedipine- and 1 µM-tetrodotoxin-containing solution in the untreated sample (CTRL) (**a**), and in the presence of 20 nM ADPN (**b**). The representative (**c**) I–V plot related to the two conditions is shown in (**a**,**b**) (open triangles, CTRL; open squares, ADPN). (**d**) Representative K^+^ current traces were obtained from a different untreated SMC (CTRL) in the presence of guanylate cyclase inhibitor ODQ 1 µM (**e**) and in the presence of ODQ + ADPN (**f**). The representative (**g**) I–V plot related to the three conditions is shown in (**e**–**f**) (open triangles, CTRL; open circles, ODQ; filled squares, ODQ + ADPN). All of the current values are normalized to cell capacitance (Cm) and are indicated in pA/pF. (**h**) I–V plots related to all of the experiments done in the four different conditions are here reported together for a better comparison: CTRL (open up triangles, *n* = 6); ADPN (open squares, *n* = 5); ODQ (open circles, *n* = 4); ADPN in the presence of ODQ (ODQ + ADPN, filled squares, *n* = 4). Data are expressed as mean ± SD. * *p* < 0.05 vs. CTRL; # *p* < 0.05 vs. ADPN. No statistically significant differences were observed between CTRL, ODQ, and ODQ + ADPN (one-way ANOVA with Bonferroni’s correction; *p* > 0.05).

**Figure 4 ijms-24-01082-f004:**
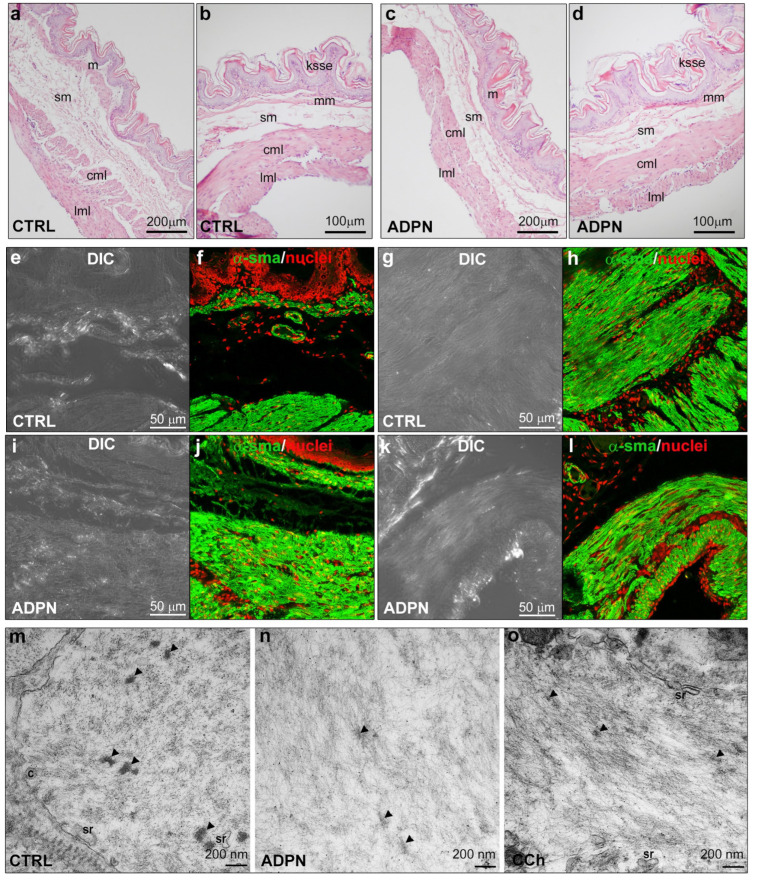
Morphological analyses. (**a**–**d**) Representative light microscopy images of sections of formalin-fixed and paraffin-embedded full-thickness strips from untreated control (CTRL) and adiponectin (ADPN)-treated murine gastric fundus stained with hematoxylin and eosin (cml = circular muscle layer, ksse = keratinized stratified squamous epithelium, lml = longitudinal muscle layer, m = mucosa, mm = muscularis mucosa, and sm = submucosa). Representative (**e**,**g**,**i**,**k**) differential interference contrast images (DIC, grey) and (**f**,**h**,**j**,**l**) relative confocal immunofluorescence images showing the expressions and distribution of α-smooth muscle actin (sma) (green). Nuclei are counterstained in red with propidium iodide (PI). Note that immunostaining of α-sma is along the muscularis mucosa, in the blood vessel wall in the submucosa (**f**,**j**), and in the circular and longitudinal muscle layers (**h**,**l**). (**m**–**o**) Representative ultrastructural transmission electron microscopy (TEM) images of the peripheral cytoplasm of longitudinally sectioned smooth muscle cells (SMCs) of the gastric fundus muscle layer from the CTRL, ADPN-treated strip, and strip treated with carbachol (CCh) to stimulate contraction (positive control). (**m**) A CTRL cell shows several bundles of irregularly arranged contractile filaments intermingled with dense bodies (arrowheads). (**n**) An ADPN-treated cell shows a slightly looser network of contractile filaments as compared with the CTRL. (**o**) A cell treated with CCh shows a denser network of contractile filaments as compared with the control. No substantial differences in the amount and morphology of plasma membrane caveolae (c) or sarcoplasmic reticulum profiles (sr) were observed among the different experimental samples (×60,000).

**Figure 5 ijms-24-01082-f005:**
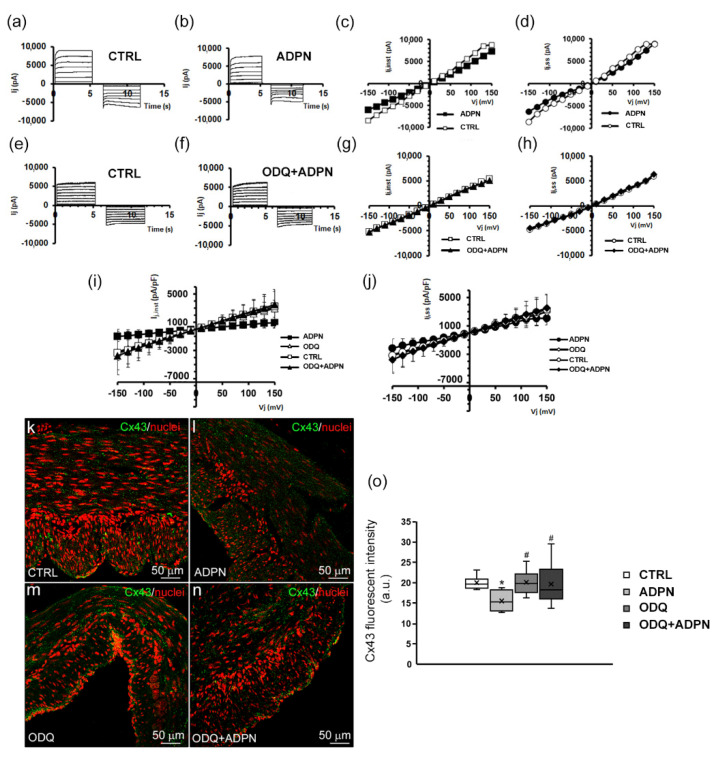
Adiponectin’s (ADPN) effect on trans-junctional currents (Ij) in smooth muscle cells (SMCs) of the gastric fundus. (**a**,**b**) Representative original trans-junctional currents tracings, Ij (in pA), recorded in response to a bipolar pulse protocol applied to a gastric SMC before (CTRL) and after an ADPN addition to the bath solution (final concentration 20 nM). (**c**) Voltage dependence of the trans-junctional instantaneous current (Ij,inst in pA) recorded from the cell analyzed in (**a**,**b**) in the absence (open squares, CTRL) or presence of ADPN (filled squares). (**d**) Voltage dependence of the trans-junctional steady state current (Ij,ss in pA) recorded from the same cell (open circles, CTRL; filled circles, ADPN). (**e**,**f**) Representative original trans-junctional currents tracings, Ij (in pA) recorded from a different gastric SMC, before (CTRL) and after ADPN addition in the concomitant presence of guanylate cyclase inhibitor ODQ (ODQ + ADPN) to the bath solution (final concentrations: ODQ 1 µM and ADPN 20 nM). (**g**) Voltage dependence of the trans-junctional instantaneous current (Ij,inst in pA) recorded from the cell analyzed in (**e**,**f**) in CTRL (open squares) or in the presence of ODQ + ADPN (filled triangles). (**h**) Voltage dependence of the trans-junctional steady state current (Ij,ss in pA) recorded from the same cell (open circles, CTRL; filled diamonds, ODQ + ADPN). (**i**,**j**) Mean trans-junctional current values normalized for Cm (in pA/pF), recorded from all of the SMCs analyzed (*n* = 4 for CTRL and ADPN; *n* = 3 for ODQ and ODQ + ADPN), were plotted vs. the trans-junctional voltage, Vj. Both the overall Ij,inst–Vj and the Ij,ss–Vj plots are not statistically different from each other in the different conditions (*p* > 0.05, one-way ANOVA). Data are expressed as mean ± SD. (**k**–**o**) Representative confocal immunofluorescence images of Connexin (Cx)43 expression (green) in the four indicated experimental conditions. Nuclei are counterstained in red with PI. (**o**) Quantitative analyses of Cx43 fluorescence intensity (a.u.) performed on digitized images. * *p* < 0.05 vs. CTRL; # *p* < 0.05 vs. ADPN (one-way ANOVA with Bonferroni’s post hoc test).

**Table 1 ijms-24-01082-t001:** Effects of different treatments on passive properties of gastric fundus smooth muscle cells (SMCs).

	CTRL	ADPN	ODQ	ODQ + ADPN
RMP (mV)	−30.9 ± 3.3	−44.0 ± 11.3 *	−29.1 ± 5.7 #	−28.5 ± 6.4 #
Cm (pF)	17.0 ± 9.8	24.7 ± 4.2 *	16.4 ± 8.6	15.8 ± 8.1

Data are as mean ± SD. * *p* < 0.05 vs. CTRL; # *p* < 0.05 vs. ADPN (one-way ANOVA with Bonferroni’s correction).

**Table 2 ijms-24-01082-t002:** Adiponectin (ADPN) levels in patients and healthy participants.

	AN (*n* = 34)	BN (*n* = 14)	BED (*n* = 14)	HC (*n* = 41)	F
ADPN (µg/mL)	11.95 ± 4.91	12.08 ± 3.30	13.30 ± 5.68	11.37 ± 4.80	0.51

AN, Anorexia Nervosa; BED, Binge-Eating Disorder; BN, Bulimia Nervosa; HC, Healthy Controls. The comparison between groups was performed using ANOVA (differences were not statistically significant; *n* = the number of subjects).

**Table 3 ijms-24-01082-t003:** Psychopathological and behavioral correlates of adiponectin (ADPN) levels in Eating Disorder (ED) patients.

	Age- and BMI-Adjusted β
EDE-Q dietary restraint	−0.10
EDE-Q eating concern	−0.08
EDE-Q weight concern	−0.09
EDE-Q shape concern	−0.09
EDE-Q total score	−0.10
Total overeating episodes	−0.01
Objective binge-eating episodes	0.08
Subjective binge-eating episodes	−0.12
Self-induced vomiting	0.03
Laxatives	−0.00
Diuretics	0.18
Compensatory exercise episodes	0.14
EES anger	0.05
EES anxiety	0.01
EES depression	0.03
EES total score	0.03
SCL-90-R GSI	−0.12

For every row variable, an age- and BMI-adjusted linear regression model was performed (ADPN as the independent variable). BMI: Body Mass Index; EDE-Q: Eating Disorder Examination Questionnaire; EES: Emotional Eating Scale; SCL-90-R GSI: Symptom Checklist-90 Revised Global Severity Index. The β-values are reported (no significant correlations are observed).

## Data Availability

Data are available from the corresponding author upon reasonable request.

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
