# Peer review of "Defining the Molecular Mechanisms of the Relaxant Action of Adiponectin on Murine Gastric Fundus Smooth Muscle: Potential Translational Perspectives on Eating Disorder Management"

_ijms, 2023, doi:10.3390/ijms24021082_

Round 1

Reviewer 1 Report

The current study supports the adiponectin (ADPN) relaxant action on gastric smooth muscle as a whole and on individual smooth muscle cells. Authors concluded that they clarified the molecular mechanisms by which ADPN exerts an inhibitory effect namely overall smooth muscle cell membrane pro-relaxant effects, with mild modifications of contractile apparatus and a slight inhibitory effect on gap junctions, all mediated by the ADPN/nitric oxide/guanylate cyclase pathway. In addition, the authors assumed the possibility of using ADPN as a biomarker of eating disorders and conducted a parallel clinical trial involving patients with eating disorders and healthy controls to assess ADPN dosage and psychopathological tests.

The preclinical research was carefully designed and has been performed by morphofunctional analyses on the murine stomach to identify cell signaling pathways engaged by ADPN. This part of the manuscript is clear, relevant to the field, and presented in a well-structured manner.

  1. Lines 74-76 The aim of this research was to further investigate the molecular mechanisms of action of ADPN on gastric fundus relaxation and to explore whether it could be considered a possible biological marker for eating disorders” – Preclinical studies were performed on mouse tissues. Obtaining the results of in-depth identification of the mechanisms of action of ADPN and transferring the animal model directly to the expected effects in a clinical trial seems insufficiently justified. The above formulation of the purpose of the study may suggest that preclinical studies were performed on several animal models or on human tissues. Mice have served and will continue to serve as valuable models for the study of basic biological processes, but the authors haven’t mentioned any known concerns regarding the reliability of relying on most animal data to predict whether an intervention will have a favorable clinical benefit-risk balance in humans. (Ioannidis, J. P. A. (2012). Extrapolating from Animals to Humans. Sci. Transl. Med. 4. doi:10.1126/scitranslmed.3004631.)
  2. The above argument also applies to the title of the manuscript, which may suggest that the research was performed on human tissues and on patients with eating disorders, and not on selected psychiatric conditions accompanied by eating disorders.
  3. Line 51 “ findings in mice suggested that ADPN, whose plasma levels usually rise with weight loss or diet restriction” with reference to a review article (Aljafary, M. A.; Al-Suhaimi, E. A. Adiponectin System (Rescue Hormone): The Missing Link between Metabolic and 600 Cardiovascular Diseases. Pharmaceutics 2022, 14, 1430. doi: 10.3390/pharmaceutics14071430.). It’s quite difficult to find an appropriate reference when 92 positions of bibliography are in that review. “However, clinical explanations of blood adiponectin concentrations should be tested according to certain factors like patient’s history of CVD, gender, hypertension age, and hemoglobin levels. Basic science research has demonstrated useful effects of adiponectin molecule on glucose balance, apoptosis, ROS stress, chronically low-grade inflammation, atherosclerosis, cardiac systolic, hypertension, etc. “ – it’s quite easy to find information that was not taken into account in the Clinical study at the Psychiatric Unit (lines 270-298 or 514- 543)
  4. The conclusions regarded the clinical part of the research as negating the working hypothesis “that ADPN may be a valid biomarker exploitable for ED management.” But only patients with anorexia nervosa; binge-eating disorder; bulimia nervosa; and healthy controls were examined.
  5. The manuscript includes 9 self-citations which is almost 20% of all citations (an excessive number of self-citations).
  6. The ethics statements and data availability statements are adequate.

Author Response

REVIEWER 1

Comments and Suggestions for Authors

The current study supports the adiponectin (ADPN) relaxant action on gastric smooth muscle as a whole and on individual smooth muscle cells. Authors concluded that they clarified the molecular mechanisms by which ADPN exerts an inhibitory effect namely overall smooth muscle cell membrane pro-relaxant effects, with mild modifications of contractile apparatus and a slight inhibitory effect on gap junctions, all mediated by the ADPN/nitric oxide/guanylate cyclase pathway. In addition, the authors assumed the possibility of using ADPN as a biomarker of eating disorders and conducted a parallel clinical trial involving patients with eating disorders and healthy controls to assess ADPN dosage and psychopathological tests.

The preclinical research was carefully designed and has been performed by morphofunctional analyses on the murine stomach to identify cell signaling pathways engaged by ADPN. This part of the manuscript is clear, relevant to the field, and presented in a well-structured manner.

  1. Lines 74-76 The aim of this research was to further investigate the molecular mechanisms of action of ADPN on gastric fundus relaxation and to explore whether it could be considered a possible biological marker for eating disorders” – Preclinical studies were performed on mouse tissues. Obtaining the results of in-depth identification of the mechanisms of action of ADPN and transferring the animal model directly to the expected effects in a clinical trial seems insufficiently justified. The above formulation of the purpose of the study may suggest that preclinical studies were performed on several animal models or on human tissues. Mice have served and will continue to serve as valuable models for the study of basic biological processes, but the authors haven’t mentioned any known concerns regarding the reliability of relying on most animal data to predict whether an intervention will have a favorable clinical benefit-risk balance in humans. (Ioannidis, J. P. A. (2012). Extrapolating from Animals to Humans. Sci. Transl. Med. 4. doi:10.1126/scitranslmed.3004631.)

ANSWER: We thank the Reviewer for this concern. Probably the way the former version of the study was presented assumed quite a big leap from the animal model to human subjects. In the revised MS we have reformulated the idea of a potential translation of  preclinical data obtained on animal models to patients, better explaining our aims and enlightening the limitations of the study. In the revised version, we added a dedicated paragraph (3.1 Study limitations and future perspectives) in Discussion. Actually, after having investigated the molecular mechanisms of action of ADP in gastric fundus relaxation in an animal model to lay the foundation for future translational studies on human subjects, we decided to perform a first, less invasive as possible analysis on blood samples from healthy subjects and patients suffering from the most common ED. Since our present data on ADPN levels seem not to show any significant correlation with the investigated ED, we may think that, for the moment, it is not worth repeating further morphofunctional investigation of ADPN effects on human gastric samples aimed to consider it a potential marker for ED.

2.The above argument also applies to the title of the manuscript, which may suggest that the research was performed on human tissues and on patients with eating disorders, and not on selected psychiatric conditions accompanied by eating disorders.

ANSWER: According to the reviewer criticism we changed the title of the manuscript. The new title is: “Defining the molecular mechanisms of the relaxant action of adiponectin on murine gastric fundus smooth muscle: potential translational perspectives  in the eating disorders management”    

3. Line 51 “ findings in mice suggested that ADPN, whose plasma levels usually rise with weight loss or diet restriction” with reference to a review article (Aljafary, M. A.; Al-Suhaimi, E. A. Adiponectin System (Rescue Hormone): The Missing Link between Metabolic and 600 Cardiovascular Diseases. Pharmaceutics 2022, 14, 1430. doi: 10.3390/pharmaceutics14071430). It’s quite difficult to find an appropriate reference when 92 positions of bibliography are in that review. “However, clinical explanations of blood adiponectin concentrations should be tested according to certain factors like patient’s history of CVD, gender, hypertension age, and hemoglobin levels. Basic science research has demonstrated useful effects of adiponectin molecule on glucose balance, apoptosis, ROS stress, chronically low-grade inflammation, atherosclerosis, cardiac systolic, hypertension, etc. “ – it’s quite easy to find information that was not taken into account in the Clinical study at the Psychiatric Unit (lines 270-298 or 514- 543)

ANSWER: In the revised version we have rewritten that sentence, which could actually sound misleading. Moreover, we cited the appropriate study that was included in the review (that is Nguyen T. M. D. (2020). Adiponectin: Role in Physiology and Pathophysiology. International journal of preventive medicine, 11, 136. https://doi.org/10.4103/ijpvm.IJPVM_193_20). Further details of the clinical study have been given in the revised version (new paragraph 3.1 of the Discussion; 4.3.3 of the Materials and Methods section).

4. The conclusions regarded the clinical part of the research as negating the working hypothesis “that ADPN may be a valid biomarker exploitable for ED management.” But only patients with anorexia nervosa; binge-eating disorder; bulimia nervosa; and healthy controls were examined.

ANSWER: Thank you. The conclusions have been rewritten according to this criticism, better specifying the object and the conclusions of the clinical study. In particular, we wrote: ... “On the other hand, data obtained in the clinical study did not reveal statistically meaningful correlations between ADPN plasma levels and any of the investigated clinical parameters in patients suffering from AN, BN and BED. Although with the limitations discussed above, our results seem to support a role for ADPN in eating behavior only in physiological conditions, while negating the working hypothesis that ADPN may be a valid biomarker exploitable for the management of the EDs considered in our study”.

5. The manuscript includes 9 self-citations which is almost 20% of all citations (an excessive number of self-citations).

ANSWER: Thank you. The reference list has been updated and increased up to 54 citations. The % of self-citations (useful for the background and methods reference) is now decreased accordingly (about 12%).

6. The ethics statements and data availability statements are adequate.

ANSWER: Thank you for your appreciation.

Reviewer 2 Report

The paper investigates the molecular mechanisms of action of ADPN on gastric fundus relaxation. It explores whether it could be considered a biological marker for eating disorders (ED). It concludes that ADPN relaxing action, namely overall smooth muscle cell membrane pro-relaxant effects, with mild modifications of contractile apparatus and a slight inhibitory effect on gap junctions, all mediated by the ADPN/nitric oxide/guanylate cyclase pathway. Even though the 27 clinical data failed to unravel a correlation between ADPN levels and ED, the clinical study nevertheless supports a role for this adipokine in physiological eating behavior. The work looks interesting but needs some adjustments.

Method:

As the authors evaluated the in vivo and ex vivo effects, a figure with the experimental design could facilitate the compression of the work.

ELISA test: please provide intra-assay variability and inter-assay variability values.

What Normality test did the authors use to determine the data distribution?

The authors did not report ethical statments for the in vivo and clinical experiments.

Results:

Provide value in table 1.

Could the authors explain why they used in figures 2 and 3 One-way ANOVA, with Bonferroni's correction, while in figure 4 One-way 250 ANOVA, with Tukey's post hoc test

In the caption of figure 2, the authors report "p<0.05 vs. CTRL (Student's t-test); #p<0.05 vs. ADPN" after statistically significant differences were observed between CTRL, ODQ, and ODQ+ADPN (One-way 162 ANOVA with Bonferroni's correction; p>0.05).

The authors found that: "plasma levels in patients with AN8 did not correlate with BMI, nor were differences detected in respect with the healthy  controls."

The authors could further discuss these results and suggest possible hypotheses for the results negate the working hypothesis that ADPN may be a valid biomarker exploitable for ED management.

In conclusion, I would appreciate a summary of the most important medically relevant key points.

Authors should add a list of abbreviations that might improve it for readers.

Author Response

REVIEWER 2

Comments and Suggestions for Authors

The paper investigates the molecular mechanisms of action of ADPN on gastric fundus relaxation. It explores whether it could be considered a biological marker for eating disorders (ED). It concludes that ADPN relaxing action, namely overall smooth muscle cell membrane pro-relaxant effects, with mild modifications of contractile apparatus and a slight inhibitory effect on gap junctions, all mediated by the ADPN/nitric oxide/guanylate cyclase pathway. Even though the 27 clinical data failed to unravel a correlation between ADPN levels and ED, the clinical study nevertheless supports a role for this adipokine in physiological eating behavior. The work looks interesting but needs some adjustments.

Method:

QUERY 1)- As the authors evaluated the in vivo and ex vivo effects, a figure with the experimental design could facilitate the comprehension of the work.

ANSWER: Thank you. We added the scheme of the experimental design in the revised version of the MS as suggested and now it appears as a new Figure 1. Accordingly, we also included a graphical abstract. Figures presented in the former version of the MS have been renumbered.

QUERY 2)-ELISA test: please provide intra-assay variability and inter-assay variability values.

ANSWER: This information was added in the Results section (2.2.Clinical study) : “The intra-assay variability was 1.80 %CV (coefficient of variation), whereas the overall inter-assay variability was 8.89 %CV”

QUERY 3)-What Normality test did the authors use to determine the data distribution?

ANSWER: We used Shapiro-Wilk test considering that we have a small sample size. This information is reported in the Materials and Methods section in paragraph: 4.3.1. Statistical analysis of electrophysiological data.

QUERY 4)-The authors did not report ethical statments for the in vivo and clinical experiments.

ANSWER: At the end of the paper, after the Funding section, we wrote the following statement: “Institutional Review Board Statement: The animal study followed the guidelines of the European Communities Council Directive 2010/63/UE and the recommendations for the care and use of laboratory animals approved by the Animal Care Committee of the University of Florence, Italy (Authorization from the Italian Ministry of Health nr. 787/2016-PR and 0DD9B.N.ZB6/2020 to M.C.B.). The clinical study involving humans was conducted in accordance with the Declaration of Helsinki, and approved by the Ethics Committee of Area Vasta Centro, reference number OSS.14.162 (date of approval 10/11/2014).

Results:

QUERY 5) Provide value in table 1.

ANSWER: We have provided values of electrophysiological data in Table 1 as suggested. As a consequence, the other two Tables shown in the former version of the manuscript have been renumbered and now they are Table 2 and Table 3.

QUERY 6)-Could the authors explain why they used in figures 2 and 3 One-way ANOVA, with Bonferroni's correction, while in figure 4 One-way ANOVA, with Tukey's post hoc test. 

ANSWER: We thank the reviewer for the query. In the revised version of the MS also the analysis shown in ex-Figure 4 was performed with One-way ANOVA with Bonferroni's correction. The outcomes gave us information comparable to those obtained with Tukey’s post hoc test. This information is given also in the Materials and Methods section (para 4.3.2 ).

QUERY 7) In the caption of Figure 2, the authors report "p<0.05 vs. CTRL (Student's t-test); #p<0.05 vs. ADPN" after statistically significant differences were observed between CTRL, ODQ, and ODQ+ADPN (One-way 162 ANOVA with Bonferroni's correction; p>0.05).

ANSWER: Thank you. The caption of ex-Figure 2 has been corrected in the revised version eliminating the wording “Student’s t test”. “Student's t-test” was just a typo deriving from a former graphical representation of only two sets of data (CTRL and ADPN). Since Panel h now actually shows four sets of data at the same time, we indeed compared them by One-way ANOVA with Bonferroni's correction.

QUERY 8)- The authors found that: "plasma levels in patients with AN did not correlate with BMI, nor were differences detected in respect with the healthy controls."The authors could further discuss these results and suggest possible hypotheses for the results negate the working hypothesis that ADPN may be a valid biomarker exploitable for ED management.

ANSWER:  In the revised version we have given a possible explanation on the bases of previous literature and proposed some limitations that could have concealed the results of the study (Discussion, paragraph 3.1). “ADPN blood levels generally fluctuate in ED and can depend on the feeding status (Tang et al., 2021). Importantly, in AN they can be restored by weight recovery, even with little increase of BMI (Tagami et al., 2004). Since it has been observed that subjects having a thin constitution have a predisposition to show higher ADPN levels related to their lower BMI, nutrition can indeed determine the reduction of this adipokine as they otherwise might have shown hyperadiponectinemia according to BMI (Tagami et al., 2004). A further limitation of the study could be that, although all analyses on clinical samples were BMI-adjusted, the evaluation of BMI may not reflect the actual fat mass quantity and therefore the actual amount of adipose tissue able to produce ADPN. Some other techniques such bioelectrical impedance analysis of body fat percentage could be used for a more reliable evaluation of body fat mass (Lai et al., 2022).”

QUERY 9)-In conclusion, I would appreciate a summary of the most important medically relevant key points.  

ANSWER: We added a paragraph of Conclusions to highlight the most medically relevant key points that are: 1) not statistically meaningful correlations between ADPN plasma levels and any of the investigated clinical parameters in patients suffering from ED; 2) a likely  role for ADPN in  physiological eating behavior; 3) at the moment, ADPN may not be a valid biomarker exploitable for the management of the EDs; 4) our findings on the molecular mechanisms of action of ADPN can help develop new effective diagnostics and therapeutics of gastric motility disturbances/impaired gastric accomodation.

QUERY 10)-Authors should add a list of abbreviations that might improve it for readers.

ANSWER: A list of abbreviations has been added in the revised version at the end of the text.